# Snap evaporation of droplets on smooth topographies

Gary G. Wells[1], Élfego Ruiz-Gutiérrez [1], Youen Le Lirzin[1,2], Anthony Nourry[1,2], Bethany V. Orme[1], Marc Pradas[3] & Rodrigo Ledesma-Aguilar [1]

Droplet evaporation on solid surfaces is important in many applications including printing, micro-patterning and cooling. While seemingly simple, the configuration of evaporating droplets on solids is difficult to predict and control. This is because evaporation typically proceeds as a "stick-slip" sequence—a combination of pinning and de-pinning events dominated by static friction or "pinning", caused by microscopic surface roughness. Here we show how smooth, pinning-free, solid surfaces of non-planar topography promote a different process called snap evaporation. During snap evaporation a droplet follows a reproducible sequence of configurations, consisting of a quasi-static phase-change controlled by mass diffusion interrupted by out-of-equilibrium snaps. Snaps are triggered by bifurcations of the equilibrium droplet shape mediated by the underlying non-planar solid. Because the evolution of droplets during snap evaporation is controlled by a smooth topography, and not by surface roughness, our ideas can inspire programmable surfaces that manage liquids in heat- and mass-transfer applications.

[1] Smart Materials & Surfaces Laboratory, Faculty of Engineering and Environment, Northumbria University, Ellison Place, Newcastle upon, Tyne NE1 8ST, UK. [2] Institut Universitaire de Technologie de Lannion, Rue Édouard Branly, 22300 Lannion, France. [3] School of Mathematics and Statistics, The Open University, Milton, Keynes MK7 6AA, UK. These authors contributed equally: Gary G. Wells, Élfego Ruiz-Gutiérrez. Correspondence and requests for materials should be addressed to R.L.-A. (email: rodrigo.ledesma@northumbria.ac.uk)

The configuration of evaporating droplets on a solid topography—e.g., their shape and location—is important for a broad range of applications. For example, in microcontact printing (a soft-lithography etching technique), an "ink" made of a polymer-solvent mixture is applied to a surface of designed topography and allowed to evaporate; the dry polymer residue is then printed onto a target surface, leaving a negative pattern of the original topography that can be used to replicate structures en masse from a single master template[1]. In immersion lithography (a widely used technique for integrated circuits manufacturing), a liquid water bridge is used to increase the precision of a UV light source for curing a target resin; an undesired side effect is the formation of droplets on the cured resin, which, upon evaporation, leave "water marks" that can spoil pattern features[2]. Spatiotemporal control of evaporating liquids is also attractive, as in edge lithography, where an ink droplet is left to evaporate on a hydrophobic patch; here, the low surface energy of the patch induces a transient dewetting process, which guides the ink residue to form edge patterns[3]. Finally, droplet evaporation is very important in heat-transfer applications[4]. For instance, a recently reported jumping-drop technique exploits super-hydrophic surfaces to induce the motion of evaporating-condensing droplets for "hotspot cooling", and is a promising heat-management technique in microelectronics[5]. These applications, however, depend on the control over the position and shape of the liquid, and this is often limited by solid–liquid–gas interactions occurring at the droplet's edge.

Since it was first proposed by Picknett and Bexon in the 1970s[6], the so-called stick-slip model has remained a canonical framework to explain the evaporation of droplets on solid surfaces. During stick-slip, a droplet alternates between two ideal "modes" as its volume is reduced: a slip mode, where the droplet edge smoothly retracts from the solid, and a stick mode, where the edge remains pinned to it. The slip mode (also called constant-contact-angle mode) is a diffusion-dominated process, where small gradients in the humidity over the surface of the droplet only drive weak hydrodynamic flows. As a consequence, the liquid and gas phases remain at rest while the interface smoothly reduces in size following the law $R(t)^2 \sim t_e - t$, where $R$ is the base radius of the droplet, $t$ is time and $t_e$ is the time at which the droplet completely evaporates[7]. The stick mode, on the other hand, involves a static contact line, i.e., $R(t) = $ const. Because of this geometrical constraint a radial flow develops to make up for the mass lost at the pinned edge upon evaporation[8]; any solid particles suspended in the liquid drift to the edge, and this is the mechanism responsible for the familiar ring-like stains left behind by coffee drops[9,10].

It is widely accepted that transitions from stick to slip, called de-pinning events, are activation processes[11]. Microscopically, a solid surface has chemical[12] or topographical[13] defects that impose a static energy barrier, hampering the translational motion of the contact line. As a consequence, an evaporating droplet with a pinned contact line stores surface energy as its volume is reduced. This proceeds until the energy barrier due to pinning is overcome, the contact line depins, and the motion of the interface is restored[14–17].

So far, the widespread conception has been that contact-line pinning caused by microscopic surface roughness dominates the evolution of evaporating droplets. Such a fundamental aspect poses severe limitations to predict and control the configuration of a droplet upon evaporation.

Here we show that droplets evaporating on a smooth—but non-flat—solid surface exhibit a different mode of evaporation: instead of pinning the droplet in an uncontrolled manner, the underlying smooth topography promotes a reproducible sequence of well-defined droplet configurations paced by dynamic "snap"

events. Such a snap mode of evaporation has the unique advantages of precise predictability and controllability over the shape and location of the droplet as it evaporates, making it useful for applications that need efficient mass and heat transfer at sub-millimetre scales.

## Results

**Evaporation on ultra-smooth liquid-impregnated rough surfaces**. We investigated the response of evaporating water droplets to a smooth topography using Lubricant-Impregnated Rough surfaces (LIRs). LIRs are solid surfaces of arbitrary shape that are first treated with a super-hydrophobic nano-coating to create a rough, water-repellant, surface, and then impregnated by a lubricant oil (see Supplementary Notes 1–3 for fabrication details). The oil creates a thin lubricating layer, of thickness $\ell \sim$ 10 μm, that covers the solid roughness, creating an ultra-smooth surface (for instance, on a tilted flat LIRs, a water droplet has a sliding angle below 1°).

We first tested the evaporation of a water droplet on a flat LIR surface (Fig. 1a). We found that the squared-base radius of the droplet decreases linearly with time, while the apparent contact angle, $\theta_a$ (measured relative to the horizontal), decreases smoothly due to the effect of a wetting lubricant ridge located at the base of the droplet[18,19] (Fig. 1b). Such kinematics, which persists for up to ~80% of the evaporation time, indicate that contact-line pinning effects are negligible[20].

We then carried out experiments of droplets evaporating on a wavy LIR surface (Fig. 1c). We placed an 80–μL droplet on a surface of wavelength $\lambda = 2$ mm and amplitude $\epsilon = 0.2$ mm, and left it to evaporate under room temperature and humidity conditions. The droplet quickly settled to adopt a symmetric shape (on a plane parallel to the wave) with its left and right edges lying close to the peaks of the topography (see panel 1 in Fig. 1c). As evaporation took place, we tracked the base radius of the droplet (a measure of the droplet's contact area) and the apparent contact angle (measured relative to the horizontal and at the intersection of the droplet's surface with the sinusoidal LIR surface (inset of Fig. 1d)). Contrary to the smooth evaporation observed on a flat surface, we found that the non-flat topography promotes a different evaporation kinematics. Initially, evaporation results in a slow retraction of the contact lines from the peaks of the topography, as shown by the continuous decrease of the base radius observed in Fig. 1d. Such kinematics is interrupted when, suddenly, one of the edges of the drop "snaps" by retracting to the adjacent peak (see images 2–3 in Fig. 1 and Supplementary Movie 1). The duration of a snap event is very short compared to the evaporation time of the droplet. Therefore, a snap appears as a discontinuous change in the lateral base radius and the apparent contact angle in the timescale of our experiments (Figs. 1d and 1e). Once a snap has occurred, the droplet continues to evaporate in a smooth manner (again, with the contact lines slowly retracting from the peaks of the wave) until another snap event is triggered. We found that sequential snaps undergone by the same droplet can be triggered at either of its edges (compare, e.g., images 3–4 and 4–5 in Fig. 1c). In addition, we found that the apparent contact angles of the left and right edges remain remarkably close to each other during the whole evaporation process. These features suggest that the smooth surface provided by the lubricant layer of the LIR surfaces eliminates contact-line pinning, and, therefore, that snaps are not de-pinning events. This contrasts with the stick-jump dynamics observed for droplets evaporating on periodic micro-patterned super-hydrophobic surfaces, where pinning effects dominate[21–23].

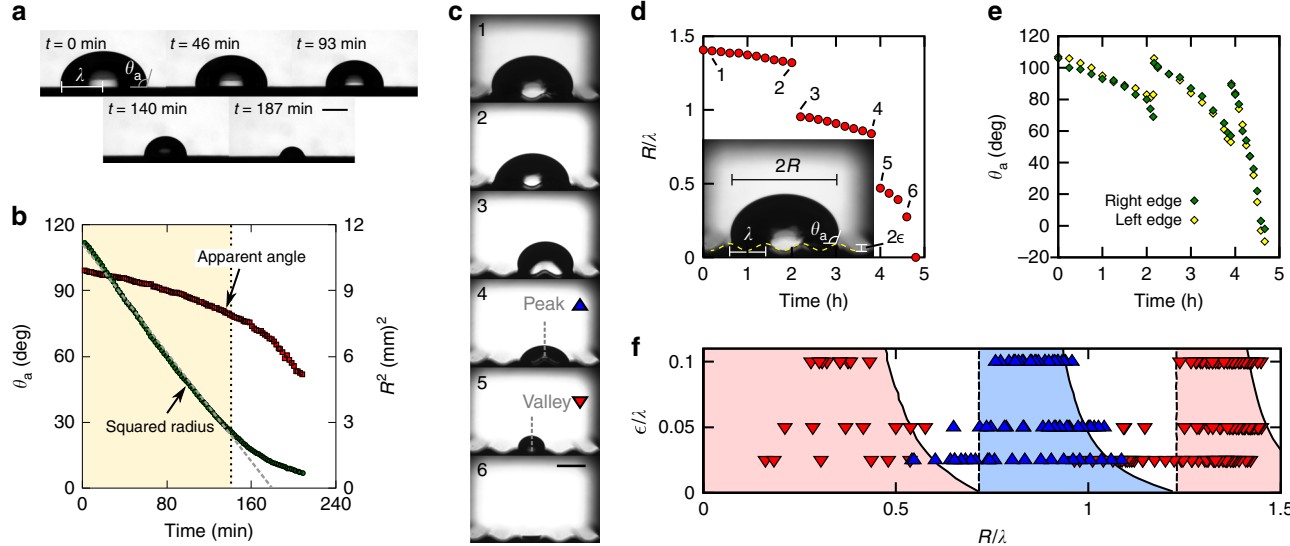

**Fig. 1** Snap evaporation on a smooth wavy surface. **a** Time-lapsed images of an 80-μL water droplet evaporating on a flat LIR surface. **b** Evaporation kinematics on a flat surface, characterised by the squared-base radius ($R^2$) and apparent contact angle ($\theta_a$) as functions of time. The diffusion-dominated, pinning-free regime is indicated by the yellow-shaded region. **c** Time-lapsed images of an 80-μL water droplet evaporating on a wavy LIR surface of wavelength $\lambda = 2$ mm and amplitude $\epsilon = 200$ μm (inset). The dashed lines in images 4 and 5 indicate the droplet's mid-plane. **d, e** Evaporation kinematics on a wavy LIR surface, characterised by the base radius (**d**) and apparent contact angle (**e**) as functions of time. **f** Observed droplet configurations (peak in blue or valley in red) for different surface aspect ratios and droplet sizes. The dashed and continuous lines that bound the shaded regions correspond to the boundary lines of each configuration predicted by the bifurcation theory model (see text). The scale bars in (**a**, **c**) are equivalent to 2 mm

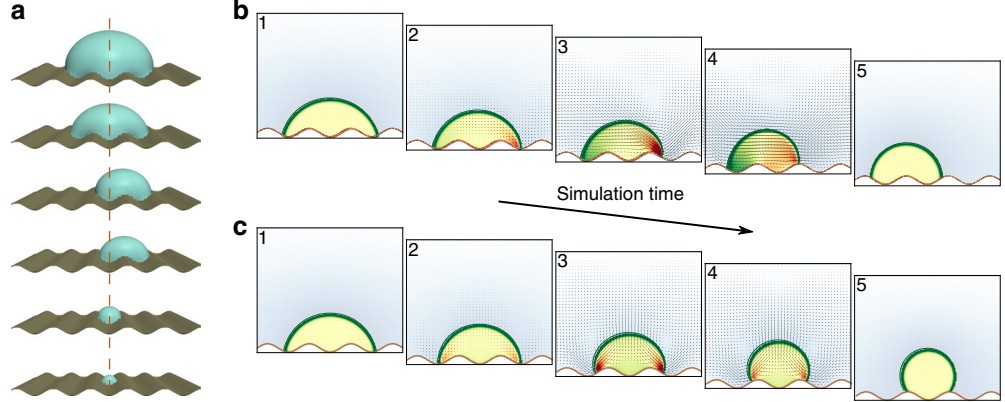

**Fig. 2** Mechanism of snap evaporation. **a** 3D Lattice–Boltzmann simulation of a droplet undergoing snap evaporation on a wavy surface. The simulation parameters match the experimental conditions of Fig. 1c. The reference midline indicates the plane of symmetry of the surface. **b, c** 2D simulations showing the dynamics of a snap event with broken (**b**) and conserved (**c**) plane symmetry. The arrows indicate the direction of the local flow pattern. The colour map outside the droplet indicates the magnitude of the local chemical potential. The colour map within the droplet indicates the magnitude of the pressure field

Between snaps, the interface shape is always symmetric about a vertical line (Fig. 1c). This symmetry is lost momentarily during a snap, while the droplet undergoes a lateral motion. As a result, the symmetry line alternates between two positions: it is either aligned with a peak of the topography or with a valley (see images 4 and 5 in Fig. 1c). Therefore, the alternation between the two configurations induces a periodic variation in the droplet's position and shape. We repeated the experiments using surfaces of different amplitude-to-wavelength ratio, $\epsilon/\lambda$, and found a highly reproducible emerging pattern: the position of the drop always alternates between peak and valley configurations, and there is a clear correlation between these configurations and the

droplet base radius which becomes increasingly marked for larger amplitudes of the topography. As illustrated in Fig. 1f, for a given droplet size and wave amplitude, it is possible to anticipate the shape of the droplet and its position relative to the solid surface.

**Lattice–Boltzmann simulations.** To better understand the mechanism of snap evaporation, we carried out numerical simulations of the coupled diffusion and hydrodynamics equations using a lattice–Boltzmann algorithm (see Supplementary Note 4 for details). In the simulations, we modelled the smooth LIR surface as a solid boundary with a small static noise in its

wettability; this allows the droplets to break the plane symmetry, but does not introduce any pinning effects. To validate our numerical model, we first considered full 3D simulations of evaporating droplets including the effect of gravity (Fig. 2a), which are in very good agreement with the sequence of droplet configurations observed in the experiments (cf. Fig. 1c). Normalising the simulation and experimental time sequences by the total evaporation time, confirms that peak and valley states always occur over the same specific ranges in the droplet base radius, and implies that the effect of the evaporation rate is purely kinematic (see Supplementary Movie 2).

Next, we carried out simulations of gravity-free 2D droplets; we found the same evaporation sequences, ruling out gravitational and 3D effects on the snap events (Fig. 2b and Supplementary Movie 3). Instead, a closer examination of the flow profiles, characterised by the velocity and pressure fields, reveals that the slow evolution of the droplets (when on a peak or a valley) is controlled by mass diffusion in the gas due to evaporation (images 1 and 5 in Fig. 2b). This situation changes when the contact lines approach the valleys of the topography. At such points, the Laplace pressure inside the droplet builds up near one of the contact lines and drives a capillary flow towards the opposite edge, triggering a snap (image sequence 2–4 in Fig. 2b). We then removed the noise from the simulations, which forces the droplet to keep the plane symmetry. Surprisingly, the droplets still undergo snaps, albeit with no translational motion (Fig. 2c and Supplementary Movie 4).

**Snap sequences and shape bifurcations.** Our numerical simulations indicate that the slow evolutions of peak and valley configurations correspond to quasi-static processes, and that during snaps the interface is out-of-equilibrium.

We expect that the 2D quasi-static droplet shapes are circular sections that intersect the solid surface with an equilibrium contact angle, $\theta_e$ (measured relative to the local surface tangent). Such interfacial shapes are indeed valid solutions of the Young–Laplace equation subject to Young's condition[13], which are the equilibrium equations for a liquid–gas interface in contact with a solid boundary[24]. On a flat surface, fixing the contact angle and the cross-sectional area of the droplet, $A$, yields a single equilibrium solution which remains invariant upon a continuous translation over the surface. On a wavy surface, corresponding to $\epsilon/\lambda > 0$, the translational invariance becomes discrete (with periodicity $\lambda$) as two symmetric equilibrium solutions appear—the peak and valley configurations. These states can be distinguished by their lateral base radius, $R$, and lateral coordinate, say $x = x_P = 0, \pm\lambda, \pm2\lambda, \ldots,$ for a peak, and $x = x_V = \pm\frac{1}{2}\lambda, \pm\frac{3}{2}\lambda, \ldots$ for a valley. This situation is maintained upon increasing the wave amplitude further, up to a critical value $(\epsilon/\lambda)^*$ where multiple circular-arc shaped equilibrium solutions (of different base radius $R$) emerge (see Supplementary Note 5). The critical amplitude thus corresponds to the onset of a cusp bifurcation[25,26]. For example, for $\epsilon/\lambda = 0.1$ and $A/\lambda^2 = 4$ there is one valley configuration and three distinct peak configurations (Fig. 3a). In addition to symmetric states, one also finds asymmetric solutions; for instance, in Fig. 3a there are four non-symmetric solutions formed by two pairs of mirror images located at intermediate positions between peaks and valleys.

The multiplicity of symmetric solutions above the critical amplitude is clearly manifested in the functional relation between $R$ and $A$, which is not bijective (see Fig. 3b). In fact, the structure of $R(A)$ curves for peak and valley states implies that reducing $A$ (e.g., due to evaporation) eventually leads to a fold where available equilibrium solutions of equivalent cross-sectional area

have a smaller radius (Fig. 3b). At first sight, one might expect that such a geometrical constraint dictates the fate of the droplets, and that snaps are triggered whenever $A$ reaches the value at the fold of the curve, $A_f$.

However, an analysis of the stability of the equilibrium solutions reveals a subtler picture. In the stability analysis, we compute the surface energy $F(R, x)$ of droplets of circular shape and prescribed cross-sectional area as a function of the base radius $R$ and the lateral coordinate $x$ (see Supplementary Note 5). Consider Fig. 3c–e, which show the evolution of the energy landscape as the area is decreased from an initial value $A/\lambda^2 = 4$, falling below the value at the fold $A_f/\lambda^2 \approx 3.11$ (see Fig. 3b). Initially, there are three sinks (the valley and the two peak configurations marked with solid lines in Fig. 3a), one source (the peak configuration marked with a dashed line) and four saddles (the asymmetric configurations). Now consider a droplet in the stable peak configuration of largest radius. As $A$ decreases and reaches a value $A_p/\lambda^2 \approx 3.26$, such a stable point merges with the two adjacent saddles, leaving a single saddle as a remnant. The area $A_p$ thus corresponds to the critical point of an inverted pitchfork bifurcation[25]. The structure of the energy landscape at $A = A_p$ explains the lateral migration of the droplet during snaps: at the bifurcation point, the remaining source prevents the droplet from migrating towards the remaining stable peak state; instead, the surface energy is always reduced upon a migration to the adjacent valley state. This sequence is repeated as the area is reduced further, and explains the clear alternation of experimental interface configurations.

The pitchfork bifurcation always occurs at a cross-sectional area $A_p$ larger than the area of the fold, $A_f$. As $A$ is reduced further from $A_p$, the saddle produced by the pitchfork bifurcation annihilates with the source at $A = A_f$, leaving a single sink in the peak branch (see Fig. 3e). Such a situation corresponds to a 2D saddle-node (or fold) bifurcation[25]. Indeed, one can remove the pitchfork bifurcation to observe the fold bifurcation by forcing the droplets to keep the plane symmetry at all times, explaining the symmetric snap events observed in the simulations (Fig. 2c). In the presence of lateral fluctuations, however, the effect of the pitchfork bifurcation is to "weaken" the fold bifurcation, producing only a remnant of the lost saddle-node.

A bifurcation diagram in $x$-$A$ space, shown in Fig. 3f, summarises the hierarchy of the pitchfork and saddle-node bifurcations governing the snapping behaviour of the droplets triggered by the sinusoidal topography. The pitchfork bifurcation always occurs when the contact lines approach the valleys of the topography, and thus the critical radius $R_p$ is independent of the amplitude of the surface pattern. However, the range of stable equilibria on a given branch becomes smaller on surfaces of a larger wave amplitude. This leads to the collapse of states at larger $\epsilon/\lambda$ observed in Fig. 1f. Despite overlooking the details of the 3D interface configuration, the 2D model gives a good prediction of the corresponding separatrices, which we present as overlays in Fig. 1f.

## Discussion

It is reasonable to expect that similar mechanisms underpin the stability of evaporating droplets on more complex topographies. Indeed, an "egg-box" surface leads to the alternation between well-defined "diamond" and "rectangle" droplet shapes (Fig. 4a). Here, again, the contact line tends to avoid the valleys of the topography, and thus the droplet adopts a shape whose typical width, $W$, and length, $L$, are multiples of the underlying wavelength of the surface pattern, $\lambda$. Snap events now involve a stepwise reduction of one of the droplet length scales, and thus the

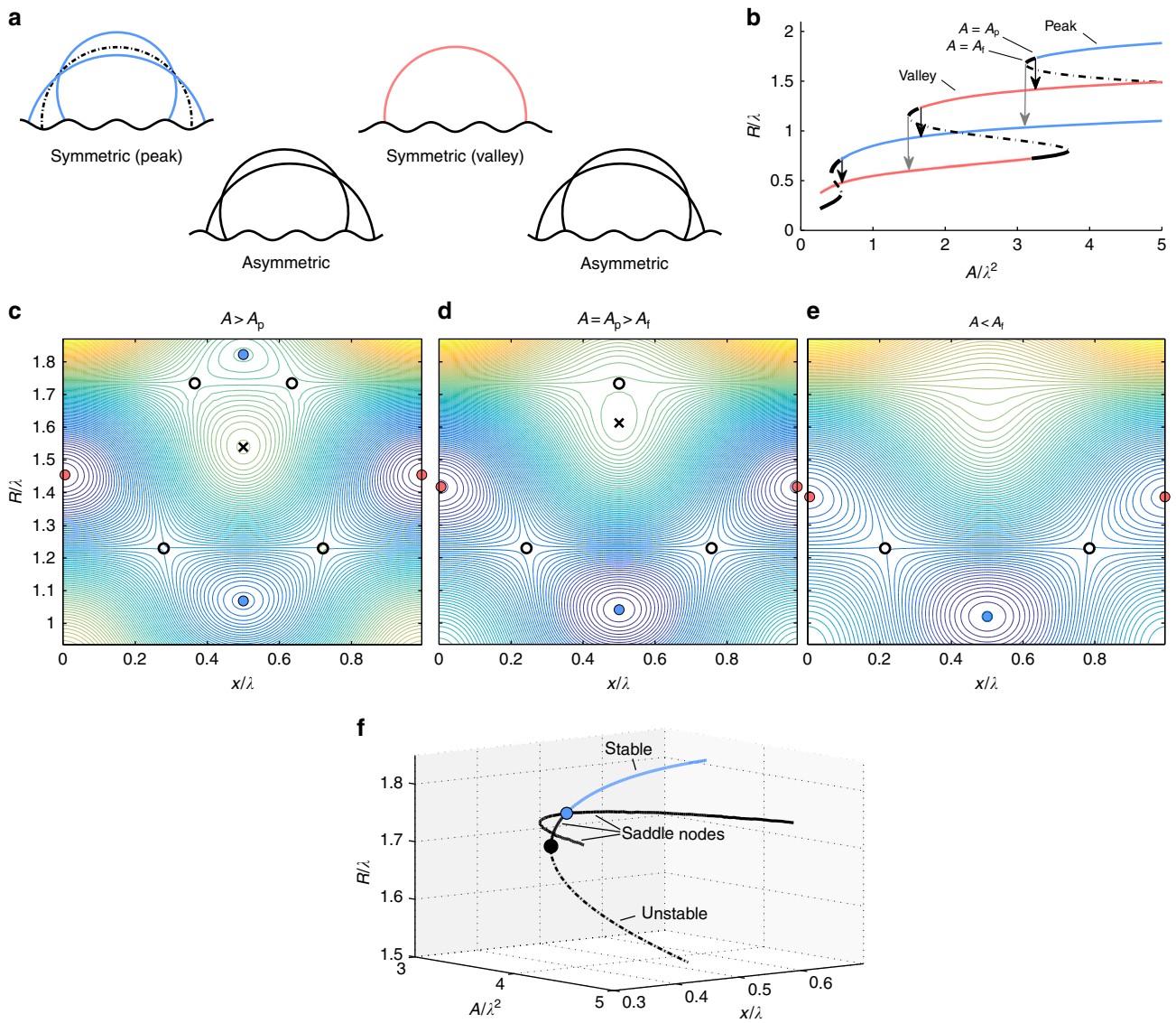

**Fig. 3** Bifurcation diagrams of equilibrium shapes of 2D droplets on a wavy surface. **a** Equilibrium droplet configurations of identical cross-sectional area $A/\lambda^2 = 4$ and intersection angle $\theta_e = 105°$ on a wavy surface of amplitude $\epsilon/\lambda = 0.1$. There are four plane-symmetric configurations: three centred at a peak and one centred at a valley. There are also four asymmetric configurations that appear as mirror-image pairs. **b** Functional relation between the lateral radius, $R$ and cross-sectional area, $A$ of plane-symmetric droplets. The blue and red branches correspond to stable equilibria on peak and valley configurations, respectively. The dashed branches correspond to unstable solutions. The solid black branches correspond to saddles, where the droplet is stable to axisymmetric perturbations, but unstable against lateral displacements along the solid surface. **c–e** Evolution of the energy landscape in $R$–$x$ space for decreasing droplet cross-sectional area. The contour lines indicate levels of the surface energy, increasing from blue to yellow. Blue and red circles correspond to stable peak and valley configurations, crosses to unstable configurations and black empty circles to saddle nodes. **c** For a $A > A_p$, a droplet in a peak state with large radius is stable to both radial and lateral displacements. **d** At $A = A_p$ the stable state merges with the two adjacent saddles (asymmetric droplet configurations), leaving a single saddle point. **e** At $A = A_f$, the saddle annihilates with the symmetric unstable state. **f** Bifurcation diagram showing stable, unstable and saddle branches for a droplet in a peak configuration. The blue point corresponds to a subcritical pitchfork bifurcation, which triggers the lateral motion of the droplet to a valley configuration (not shown). The black point corresponds to the onset of a saddle-node bifurcation, which occurs only if the droplet cannot break the plane symmetry

droplet evolves following a sequence $W \times L \approx 5\lambda \times 5\lambda \rightarrow 4\lambda \times 5\lambda \rightarrow 4\lambda \times 4\lambda\ldots$, which can be exploited to control the aspect ratio of the droplet (Fig. 4b).

Therefore, snap evaporation is a distinct mode of droplet evaporation on smooth—but topographically patterned—solid surfaces. Unlike stick-slip evaporation, the alternation of well-defined quasi-static states observed in snap evaporation is controlled by shape bifurcations of the liquid–gas interface dictated by the interplay between the underlying surface topography and the droplet volume, and not by contact-line pinning.

In our experiments, the timescale of snap events is very short compared to the evaporation time of the droplet (see Fig. 1d). The regime where the evaporation and snap timescales compete poses fundamental questions in relation to the dynamics of bifurcations (for which our experimental setup provides a useful test bed), but can also find application in situations where evaporation happens in a short timescale, such as in micro-fluidics. Our ideas can also be applied to other methods of variation of the droplet volume on smooth surfaces, such as condensation[27], mass transfer via flow rate[28], or by exploiting external fields (e.g., temperature or

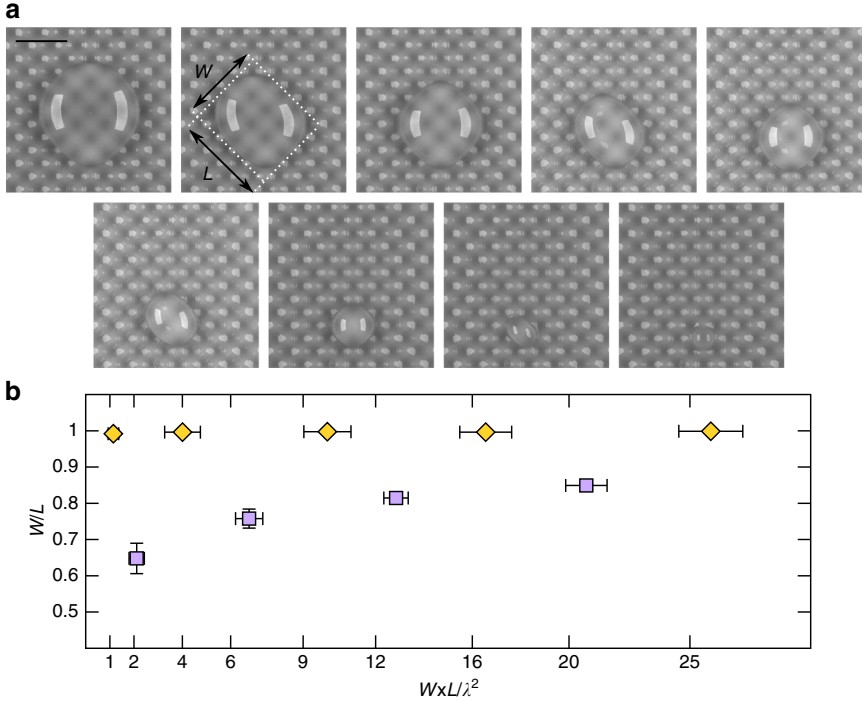

**Fig. 4** Droplet shape alternation during snap evaporation on an egg-box topography. **a** Time-lapsed sequence of an 80-μL droplet evaporating on a surface of wavelength $\lambda = 4$ mm and amplitude $\epsilon = 200$ μm. At any given time, the shape of the droplet is characterised by its width, $W$, and length, $L$. **b** Width-to-length aspect ratio as a function of the apparent contact area of the of the droplet, $W \times L$. The droplet shape alternates between 'diamond' (yellow diamonds) and 'rectangle' (purple squares) configurations with droplet size (the ticks in the $x$-axis correspond to contact areas that are multiples of the underlying grid). The scale bar in (**a**) is equivalent to 1 cm

pressure). Finally, the variation of the surface topography, either spatially through designed patterns, or dynamically via forced droplet motion[29] or actuation of flexible solids, can be used to extend these principles to achieve a better control of droplet localisation and transport mediated by snaps.

**Data availability**. The data that support the findings of this study are available from R.L.A. upon reasonable request.

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

## Acknowledgements

We would like to thank J. Sardanyés and M. Sommacal for useful discussions. E.R.G. and B.V.O. acknowledge financial support from Northumbria University via a PhD Studentship. R.L.A. and G.G.W. thank the Royal Society Research Grant Scheme for financial support (grant no. RG150470); E.R.G. and R.L.A. acknowledge support from EPSRC (grant no. EP/P024408/1). R.L.A. acknowledges support from EPSRC as a member of the UK Consortium on Mesoscale Engineering Sciences (grant no. EP/L00030X/1).

## Author contributions

M.P. and R.L.A. conceived the research. M.P., R.L.A. and G.G.W. supervised the research. G.G.W. designed the experiments. G.G.W., Y.L.L. and A.N. carried out the experiments of drop evaporation on flat and plane-wave surfaces. B.V.O. and G.G.W. carried out the experiments of drop evaporation on egg-box surfaces. G.G.W., R.L.A., Y.L.L., A.N. and B.V.O. analysed the experimental data. E.R.G. developed and carried out the simulations. M.P., R.L.A. and E.R.G. developed the theoretical model. R.L.A. and M.P. wrote the paper. E.R.G. and G.G.W. contributed equally to this work.

## Additional information

**Competing interests:** The authors declare no competing interests.

