## [Peer Review File · Nature Communications]

Reviewers' comments:

Reviewer #1 (Remarks to the Author):

This review is rather brief due to the tight deadline imposed by the journal and my numerous other refereeing commitments.

The paper reports results from experiments involving droplet evaporation on a topographically patterned liquid-infused surface. Complementary Lattice-Boltzmann simulations are also performed.

The results seem novel and of fundamental interest, although it is not clear to me how useful they will be for practical applications. For many practical applications involving droplet evaporation, it will not be possible to topographically pattern the substrate or to infuse it with a lubricant.

There are some recent papers regarding droplet motion and evaporation on substrates with topography that should be referenced. There are also some points in the text that need clarification. The paper should be revised to account for the comments below and resubmitted for further evaluation.

1. Several recent papers have developed models aimed at understanding the role of surface roughness on droplet motion and evaporation. In these works, pinning due to surface roughness is captured by having a single topographical "defect". By describing pinning in this way, the model yields results that are consistent with experimental observations. These papers seem very

relevant to the present paper and should be referenced. The papers are:

L. Espin and S. Kumar, Droplet Spreading and Absorption on Rough, Permeable Substrates, *J. Fluid Mech.* 784, 465-486 (2015).

J. Park and S. Kumar, Droplet Sliding on an Inclined Substrate with a Topographical Defect, *Langmuir* 33, 7352-7363 (2017).

T. Pham and S. Kumar, Drying of Droplets of Colloidal Suspensions on Rough Substrates, *Langmuir* 33, 10061-10076 (2017).

2. I don't understand the following claims in the paper, so revision of the text is necessary to provide further clarification.

p. 1: "The slip mode is a purely diffusive process, during which the liquid and gas phases remain at rest" Slip necessarily involves moving contact lines, so hydrodynamics must come into play at some level, even if the flows are relatively weak.

p. 1: "As a consequence, an evaporating droplet with a pinned contact line stores surface energy as its volume is reduced." This sounds like an argument from equilibrium thermodynamics being applied to explain a non-equilibrium process.

p. 3: "we modelled the smooth LIR

surface as a solid boundary with a small static noise in its wettability as a means to allow the droplets to break the plane symmetry, but without incurring in any pinning effects." Why is this static noise needed? Isn't it enough to simply simulate a droplet on a topographically patterned surface like those used in the experiments?

p. 6: "Therefore, snap evaporation is a different mode of evaporation of droplets that arises on smooth, but topographically patterned, solid surfaces." Isn't the surface used by the authors an example of a smooth but topographically patterned solid surface?

Reviewer #2 (Remarks to the Author):

The paper reports an experimental study accompanying simulations of a snap mode in droplet evaporation on a smooth but wavy lubricant Impregnated Rough surfaces. The work measures the droplet base radius and contact angle with time, and tracks the position of the droplet contact line on the valley or on the peak of the wavy structure on the substrate. The measurements show that the drop edge avoids the valley on the surface, but rapidly snaps and moves faster to the next peak. The authors perform Lattice-Boltzmann simulation to understand the mechanism for the snap evaporation mode. The topic is interesting in the area of wetting in general. However, the finding of the snap mode is not new. Almost same mode has been reported in other circumstance, although the mode is named differently, for instance, stick-jump mode or stick-slip mode with contact angle oscillation. Moreover, the manuscript has not clearly explained some important technical points. In addition to reference 15 cited in the manuscript, similar work has been reported in other papers. For example,

Rodney Marcelo do Nascimento, Cécile Cottin-Bizonne, Christophe Pirat, and Stella M. M. Ramos, Water Drop Evaporation on Mushroom-like Superhydrophobic Surfaces: Temperature Effects. *Langmuir* 2016.

Damien Debuisson, Alain Merlen, Vincent Senez, and Steve Arscott, Stick–Jump (SJ) Evaporation of Strongly Pinned Nanoliter Volume Sessile Water Droplets on Quick Drying, Micropatterned Surfaces. *Langmuir* 2016, 32, 2679–2686

We feel that, due to the lack of novelty, this work is not suitable for acceptance to *Nature Communications*. We recommend that the authors submit the manuscript to another journal, such as *Soft Matter* or *Langmuir*, after significant improvement of the manuscript.

1. In Abstract, the authors describe their substrate as ‘smooth, pinning-free, solid

surfaces of non-planar topography’. However, in the Results they state that the substrate is ‘Lubricant Impregnated Rough surfaces’. This is contradictory.

2. Figure 1 has not really shown the duration for the droplet edge on the peak or on the valley. It is essential to provide a plot relating the position of the droplet edge and time in order to support the authors’ claim that the droplet avoids the valley.

3. What about droplet height, and volume and center shift with evaporation? What is the reference plane for the apparent contact angle at peak or valley?

4. Does the evaporation rate influence the snap mode? How do the authors control the evaporation rate in the experiments?

5. The figure in Outlook and Conclusions can be moved to Results and Discussion. So the structure of the manuscript is easier to follow.

Reviewer #3 (Remarks to the Author):

Many thanks for referring this excellent paper to me. This is indeed a fantastic work with original thought and worthy of publishing in Nature Comms. The manuscript on the whole is written well but I have a few questions that should be clarified in the manuscript.

1) While I understand the concept of using cross-sectional area, A , I don't think this has been justified enough. Why not use the contact area (or contact perimeter) which directly influences pinning-depinning behaviour? Especially for a hydrophobic substrate for instance, the contact area may be small but the cross-sectional area may be large, which means that I do understand that cross-sectional area works great from experimental standpoint as it is easier to measure from images, and therefore can be used to interpret the behaviour.

2) The contact-line boundary conditions in the LB model are not explained. Am I correct in understanding that while no-slip BC are enforced on the solid-fluid interface, at the triple line the Cahn-Hilliard equation results in a Van der Waal driven mobility (because of gradient of chemical potential) of the interface as a function of the contact angle (see Ding and Spelt (JFM, 2007), Ding and Spelt (PRE, 2007) etc which are based on Jacqmin (JFM, 2000). A typical drawback of these modelling approaches have been that one can either simulate a fully pinned or a fully dynamic contact lines. The authors present interesting results which can simulate both pinning, depinning and dynamic contact line behaviour. But - no real guidance is given on the boundary conditions that are used. The authors must augment this discussion in the supplementary information.

3) The authors must explain the consequence of the boundary conditions used on depinning events that is central to this paper. What physics do they infer to enable depinning to happen?

4) How do the contact line dynamics (angle, depinning and pinning, mobility) depend on temperature/ evaporative fluxes?

5) What is/are the signature of the evaporative fluxes during the snap evaporation? Especially close to bifurcation regions.

Reviewer #1 (Remarks to the Author):

We would like to thank the Reviewer for his/her careful reading and comments on our paper. We have addressed all points by the reviewer and provide a list of answers below. We also indicate where changes have been made to the manuscript.

This review is rather brief due to the tight deadline imposed by the journal and my numerous other refereeing commitments.

The paper reports results from experiments involving droplet evaporation on a topographically patterned liquid-infused surface. Complementary Lattice-Boltzmann simulations are also performed.

The results seem novel and of fundamental interest, although it is not clear to me how useful they will be for practical applications. For many practical applications involving droplet evaporation, it will not be possible to topographically pattern the substrate or to infuse it with a lubricant.

We thank the Reviewer for recognising the novelty and fundamental interest of our paper. Several methods for patterning a solid surface in a controlled manner using different materials and at different length-scales exist; these include soft lithography, mould pressing and 3D printing. The plausibility of creating non-flat lubricant infused surfaces has also been established: for example, Park et al. (ref. 22 in the paper) have created “slippery bumps” by pressing thin aluminium sheets on 3D printed moulds and then treating them with a hydrophobic coating and a mineral oil. Furthermore, they have demonstrated their utility in fog harvesting applications. In the past few years, LiquiGlide, a spin-out company from the group of Kripa Varanasi at MIT, have successfully developed a method to treat non-curved surfaces with a lubricant-impregnated coating to create slippery food containers (www.liquiglide.com). Therefore, we believe that our ideas will appeal to industrialists working on applications that demand control over the position and shape of a liquid droplet, and where liquid-impregnated surfaces might be a suitable means of achieving a smooth surface.

There are some recent papers regarding droplet motion and evaporation on substrates with topography that should be referenced. There are also some points in the text that need clarification. The paper should be revised to account for the comments below and resubmitted for further evaluation.

1. Several recent papers have developed models aimed at understanding the role of surface roughness on droplet motion and evaporation. In these works, pinning due to surface roughness is captured by having a single topographical "defect". By describing pinning in this way, the model yields results that are consistent with experimental observations. These papers seem very relevant to the present paper and should be referenced. The papers are:

L. Espin and S. Kumar, Droplet Spreading and Absorption on Rough, Permeable Substrates, J. Fluid Mech. 784, 465-486 (2015).

J. Park and S. Kumar, Droplet Sliding on an Inclined Substrate with a Topographical Defect, Langmuir 33, 7352-7363 (2017).

T. Pham and S. Kumar, Drying of Droplets of Colloidal Suspensions on Rough Substrates, Langmuir 33, 10061-10076 (2017).

The paper by Espin and Kumar is a study of spreading on permeable surfaces, while the paper by Park and Kumar is a study of single-defect pinning of a droplet sliding down an incline. We thank the reviewer for pointing us to these papers that fall within the broad field of droplet pinning, but we feel that their relevance to the current work

on evaporation is limited. We do, however, recognise that the theoretical paper by Pham and Kumar (which puts forward a model to describe the stick-slip mode of evaporation) is timely and relevant in relation to the present work, and have included a reference to it in the introduction.

2. I don't understand the following claims in the paper, so revision of the text is necessary to provide further clarification.

p. 1: "The slip mode is a purely diffusive process, during which the liquid and gas phases remain at rest" Slip necessarily involves moving contact lines, so hydrodynamics must come into play at some level, even if the flows are relatively weak.

We thank the Reviewer for making this point. During the diffusion-dominated dynamics the shape of the interface adjusts in a quasi-static manner to the variation in volume, and so any flows decay over a short timescale relative to the evaporation time of the drop. We now make this point more clearly using the following sentence in the paper and have added a reference.

The slip mode (also called constant-contact-angle mode) is a diffusion-dominated process where small gradients in the humidity over the surface of the droplet only drive weak hydrodynamic flows.

p. 1: "As a consequence, an evaporating droplet with a pinned contact line stores surface energy as its volume is reduced." This sounds like an argument from equilibrium thermodynamics being applied to explain a non-equilibrium process.

We agree that the free energy of the droplet is an equilibrium quantity. However, we refer to the surface energy, which is a concept applicable to an out-of-equilibrium situation.

p. 3: "we modelled the smooth LIR surface as a solid boundary with a small static noise in its wettability as a means to allow the droplets to break the plane symmetry, but without incurring in any pinning effects." Why is this static noise needed? Isn't it enough to simply simulate a droplet on a topographically patterned surface like those used in the experiments?

In the experiments the droplets always break the plane symmetry as they undergo a snap. In the simulations, the absence of noise (either thermal fluctuations or small vibrations present in the experiments) leads to snaps that conserve the plane symmetry (see Fig. 2c). Therefore, we introduce the small static noise in the surface to mimic the effect of noise in the experiments, which leads to the observed experimental asymmetric snaps (see Fig 2a and 2b).

p. 6: "Therefore, snap evaporation is a different mode of evaporation of droplets that arises on smooth, but topographically patterned, solid surfaces." Isn't the surface used by the authors an example of a smooth but topographically patterned solid surface?

We think that the wording of this sentence can lead to confusion. We have replaced the sentence with the following.

Therefore, snap evaporation is a distinct mode of droplet evaporation on smooth, but topographically patterned, solid surfaces.

Reviewer #2 (Remarks to the Author):

We would like to thank the Reviewer for his/her careful reading and comments on our paper. We have addressed all points by the reviewer and provide a list of answers below. We also indicate where changes have been made to the manuscript.

The paper reports an experimental study accompanying simulations of a snap mode in droplet evaporation on a smooth but wavy lubricant Impregnated Rough surfaces. The work measures the droplet base radius and contact angle with time, and tracks the position of the droplet contact line on the valley or on the peak of the wavy structure on the substrate. The measurements show that the drop edge avoids the valley on the surface, but rapidly snaps and moves faster to the next peak. The authors perform Lattice-Boltzmann simulation to understand the mechanism for the snap evaporation mode.

We would like to clarify that our study consists of experiments, numerical simulations and a theory based on a free-energy model of the equilibrium state of the droplet on a wavy surface, which we analyse using bifurcation theory. Our theory predicts snap events and this is supported by a direct comparison to the experimental data (see Fig. 1e).

The topic is interesting in the area of wetting in general. However, the finding of the snap mode is not new. Almost same mode has been reported in other circumstance, although the mode is named differently, for instance, stick-jump mode or stick-slip mode with contact angle oscillation. Moreover, the manuscript has not clearly explained some important technical points. In addition to reference 15 cited in the manuscript, similar work has been reported in other papers. For example,

Rodney Marcelo do Nascimento, Cécile Cottin-Bizonne, Christophe Pirat, and Stella M. M. Ramos, Water Drop Evaporation on Mushroom-like Superhydrophobic Surfaces: Temperature Effects. Langmuir 2016.

Damien Debuissou, Alain Merlen, Vincent Senez, and Steve Arscott, Stick-Jump (SJ) Evaporation of Strongly Pinned Nanoliter Volume Sessile Water Droplets on Quick Drying, Micropatterned Surfaces. Langmuir 2016, 32, 2679–2686

We feel that, due to the lack of novelty, this work is not suitable for acceptance to Nature Communications. We recommend that the authors submit the manuscript to another journal, such as Soft Matter or Langmuir, after significant improvement of the manuscript.

We disagree with the Reviewer's comment on the novelty of our paper, which uses as a basis the recent papers by do Nascimento et al. and Debuissou et al.

In our experiments we observe a smooth continuous decrease of the base radius during the quasi-static evaporation of the droplet (see Fig. 1d). This is a consequence of the thin lubricant layer, which makes the surface smooth and eliminates contact-line pinning. Our simulations of drops on smooth, pinning-free surfaces allow us to back this claim, showing that the droplet undergoes the same sequence of configurations and snap events observed in our experiments. Finally, our theory, which does not include any pinning effects, quantitatively reproduces the experimental data as evidenced by Fig. 1e.

It is true that several previous studies have reported the effect of periodic microscale patterns on the evaporation of droplets, and we recognised this fact in our paper with a citation to McHale et al. as a relevant initial work (ref. 18 in the original manuscript).

The paper by do Nascimento et al. is a study of temperature effects on the evaporation of droplets on micro-patterned superhydrophobic surfaces, whilst the paper of Debuissou et al. is a study of the effect of strongly pinned contact lines (via concentric micro-patterned trenches) on droplet evaporation time. The patterning on

both of these surfaces is periodic, and, hence, it is not surprising that in both papers the authors observe oscillations in the apparent contact angle.

However, and as the authors of these papers note themselves, pinning is the determinant mechanism affecting the motion of the contact line for such micro-patterned structures. This is evidenced by Fig. 6 in do Nascimento et al. and Fig. 3A in Debuissou et al, where the base radius remains constant as the contact line is pinned, and abruptly drops once a de-pinning event is triggered.

We also note that neither paper can claim to have control over the position and shape of the droplet during evaporation; both Fig. 6 in do Nascimento et al. and Fig. 3A in Debuissou et al show that the base radius and apparent contact angle behave as random variables when plotted vs. time.

In contrast, the snaps observed in our experiments occur in a predictable manner, as shown by Fig. 1e. This is because our smooth lubricant-infused surfaces lead to a smooth energy landscape where snaps are controlled by bifurcation points; we believe that the agreement of our numerical simulations and mathematical model with our experiments strongly backs this claim.

To better explain this point we have modified the description of our experimental observations as follows:

Initially, evaporation results in a slow retraction of the contact lines from the peaks of the topography, as shown by the continuous decrease of the base radius observed in Fig. 1d. Such kinematics is interrupted when, suddenly, one of the edges of the drop snaps by retracting to the adjacent peak [see images 2-3 in Fig. 1 and Supplementary Video 1]. The duration of a snap event is very short compared to the evaporation time of the droplet. Therefore, a snap appears as a discontinuous change in the lateral base radius and the apparent contact angle vs time [Fig. 1d]. Once a snap has occurred, the droplet continues to evaporate in a smooth manner (again, with the contact lines slowly retracting from the peaks of the wave) until another snap event is triggered. We found that sequential snaps undergone by the same droplet can be triggered at either of its edges [compare, e.g., images 3-4 and 4-5 in Fig. 1c]. In addition, we found that the apparent contact angles of the left and right edges remain remarkably close to each other during the whole evaporation process. These features suggest that the smooth surface provided by the lubricant layer of the LIR surfaces eliminates contact-line pinning, and, therefore, that snaps are not de-pinning events. This contrasts with the stick-jump dynamics observed for droplets evaporating on periodic micro-patterned super-hydrophobic surfaces, where pinning effects dominate [citations to McHale et al., do Nascimento et al. and Debuissou et al.]

1. In Abstract, the authors describe their substrate as 'smooth, pinning-free, solid surfaces of non-planar topography'. However, in the Results they state that the substrate is 'Lubricant Impregnated Rough surfaces'. This is contradictory.

We believe that our description of the LIR surfaces was unfortunate and agree that it can lead to confusion. We have changed the description of our LIR surfaces to read:

The oil creates a thin lubricating layer, of thickness $l \sim 10 \mu\text{m}$, that covers the rough surface, and which renders LIRs ultra smooth.

2. Figure 1 has not really shown the duration for the droplet edge on the peak or on the valley. It is essential to provide a plot relating the position of the droplet edge and time in order to support the authors' claim that the droplet avoids the valley.

The timescale of the motion of the contact line during a snap is much shorter than the evaporation time. This is supported by Fig. 1d, where both the base radius and the

apparent contact angle show step-like changes during snap events. We think that our description of the snap process using the term “avoiding” is unnecessary, and have thus eliminated it and modified this sentence to read (see above for full paragraph):

... suddenly, one of the edges of the drop “snaps” by retracting to the adjacent peak.

3. *What about droplet height, and volume and center shift with evaporation? What is the reference plane for the apparent contact angle at peak or valley?*

4. *Does the evaporation rate influence the snap mode? How do the authors control the evaporation rate in the experiments?*

Answers to 3. and 4. In our experiments, we identify the droplet base radius and the apparent contact angle as the relevant geometrical variables to describe the evaporation of the droplets. The centre shift is reported in Fig. 1e, where we identify peak and valley configurations using different symbols, and also in our model when analysing the equilibrium configurations of the droplet (denoted x ; see Fig. 3c and 3d). The apparent contact angle is always measured relative to the horizontal and at the intersection with the droplet’s edge; we now indicate this important point in the paper for clarity:

As evaporation took place, we tracked the base radius of the droplet (a measure of the droplet’s contact area) and the apparent contact angle (measured relative to the horizontal and at the intersection of the droplet’s surface with the sinusoidal LIR surface [inset of Fig. 1d]).

Our main claim is the identification and quantification of the mechanism leading to snaps, which we achieve by means of our analytical model. The model takes the volume (the cross-sectional area in 2D) of the droplet as a parameter, and gives a quantitative prediction of the droplet configuration as shown in Fig. 1e. This supports that in the experiments the volume of the droplet changes in a quasi-static manner, and therefore the effect of the evaporation rate is unimportant (the average evaporation rate is typically $\sim 4 \text{ nL s}^{-1}$, which we now indicate in the Supplementary Information). It would be interesting to investigate the regime where evaporation and snap dynamics compete, and we now motivate this question in the outlook section of our paper.

5. The figure in Outlook and Conclusions can be moved to Results and Discussion. So the structure of the manuscript is easier to follow.

We thank the reviewer for this suggestion. We believe that the Results section should remain devoted to the analysis of our main results on planar-wave surfaces. We believe that Fig. 4 should remain in the Outlook and Conclusions section as its purpose is to illustrate the applicability of our ideas and to motivate further studies of evaporation on smooth topographies.

Reviewer #3 (Remarks to the Author):

Many thanks for referring this excellent paper to me. This is indeed a fantastic work with original thought and worthy of publishing in Nature Comms. The manuscript on the whole is written well but I have a few questions that should be clarified in the manuscript.

We would like to thank the Reviewer for his/her careful reading and positive comments on our paper. We have addressed each point raised by the reviewer and prepared a response accordingly. We have indicated where we have made changes to the manuscript.

1) While I understand the concept of using cross-sectional area, A , I don't think this has been justified enough. Why not use the contact area (or contact perimeter) which directly influences pinning-depinning behaviour? Especially for a hydrophobic substrate for instance, the contact area may be small but the cross-sectional area may be large, which means that I do understand that cross-sectional area works great from experimental standpoint as it is easier to measure from images, and therefore can be used to interpret the behaviour.

As the reviewer points out, the contact area is an important parameter. Because the base radius is a measure of the contact area, we expect that the contact area will also exhibit a step change when a snap event is triggered. We have added the following sentence to the manuscript to better justify this point.

As evaporation took place, we tracked the base radius of the droplet (a measure of the droplet's contact area) and the apparent contact angle (measured relative to the horizontal and at the intersection of the droplet's surface with the sinusoidal LIR surface [inset of Fig. 1d]).

In our mathematical model, we identified the cross-sectional area as the relevant control parameter linked to evaporation (equivalent to the volume in 3D) and the base radius as the relevant geometrical parameter that allows us to track the configuration of the droplet. This choice of variables allows us to carry out our mathematical analysis in a more straightforward manner.

2) The contact-line boundary conditions in the LB model are not explained. Am I correct in understanding that while no-slip BC are enforced on the solid-fluid interface, at the triple line the Cahn-Hilliard equation results in a Van der Waal driven mobility (because of gradient of chemical potential) of the interface as a function of the contact angle (see Ding and Spelt (JFM, 2007), Ding and Spelt (PRE, 2007) etc which are based on Jacqmin (JFM, 2000). A typical drawback of these modelling approaches have been that one can either simulate a fully pinned or a fully dynamic contact lines. The authors present interesting results which can simulate both pinning, depinning and dynamic contact line behaviour. But - no real guidance is given on the boundary conditions that are used. The authors must augment this discussion in the supplementary information.

Our Lattice-Boltzmann simulations solve the coupled Navier-Stokes and Cahn-Hilliard equations at every time-step. We impose a no-flux boundary condition at the solid surface (this is achieved using the so-called bounce-back algorithm in LB). The wetting behavior of the fluid is modeled using a Neumann boundary condition for the phase field (Cahn, 1976; Jacqmin, 2000). The combination of both features leads to:

1) A no-slip boundary condition in each of the fluids away from the contact line and

2) A mobile contact line, which arises by virtue of the gradient in chemical potential as the reviewer notes.

In the context of Lattice-Boltzmann simulations, such boundary conditions have been validated by direct comparison with the Cox-Voinov Law—a relation between the

speed of the contact line and the dynamic contact angle (Briant and Yeomans 2004; Kusumaatmaja et al., 2016)

The papers by Ding and Spelt, and by Jacqmin are based on the same idea (although use a different numerical method) and are certainly relevant to the discussion of this important aspect of the simulations.

We have added this discussion to the Supplementary Information as suggested by the reviewer.

3) The authors must explain the consequence of the boundary conditions used on depinning events that is central to this paper. What physics do they infer to enable depinning to happen?

Our simulations clearly show that the evaporation process occurs as an alternation between two different regimes. The droplet spends most of the time undergoing a quasi-static evaporation, where the contact lines remain close to the peaks of the surface. We do not impose this condition artificially; rather, it arises because the interface has enough time to adjust to an equilibrium shape (satisfying the equilibrium contact angle with the solid). This is shown in panels 1 and 5 of Figs 2b and 2c. The second regime corresponds to a snap event, where the droplet is not equilibrium and moves from an unstable configuration. We describe the dynamics by discussing the pressure and velocity fields (see panels 2-4 in Figs 2b and 2c). During this stage, the boundary conditions drive the contact lines to move by virtue of a difference in chemical potential at the level of the contact line, which is opposed by viscous friction.

We have expanded the discussion of the role of the boundary conditions imposed in the simulations in the supplementary information.

4) How do the contact line dynamics (angle, depinning and pinning, mobility) depend on temperature/ evaporative fluxes?

5) What is/are the signature of the evaporative fluxes during the snap evaporation? Especially close to bifurcation regions.

Answer to points 4) and 5): In our experiments, the timescale of the contact line motion during snap events is very short compared to the timescale of evaporation. As shown by our numerical simulations, the droplet remains in a quasi-static state before a snap event, and this implies that the effect of the evaporation rate is purely kinematic. Therefore, the contact lines simply adjust to the new equilibrium configuration by satisfying the local contact angle relative to the wavy surface as discussed in point 3) above.

Based on this information, our mathematical model takes the volume of the droplet as a parameter (i.e., we eliminate the time dependence), and gives a quantitative prediction of the droplet configurations as shown in Fig. 1e. This supports that in the experiments the volume of the droplet changes in a quasi-static manner, and therefore the effect of the evaporation rate is unimportant. We expect that this regime holds as long as the timescale of motion of the contact lines during snaps remains short relative to the timescale of evaporation. It would be interesting to investigate the regime where evaporation and snap dynamics compete, a situation that can be achieved by increasing the temperature to augment the evaporation rate, and we now motivate this question in the outlook section of our paper.

REVIEWERS' COMMENTS:

Reviewer #1 (Remarks to the Author):

The authors have done a reasonable job addressing my comments. I think the paper can now be accepted.

Reviewer #2 (Remarks to the Author):

The authors have confirmed that the substrates in their experiments are 'Lubricant Impregnated Rough surfaces'. The original misleading information in abstract remains unchanged in the revised manuscript. The entire work is based on the point of 'pinning-free' substrate. The authors must define clearly what 'pinning' means, as the edge of drop on a lubricant film follows Neumann's triangle related to the surface energies. How is their situation compared to the contact line pinning reported in this work? Droplet mobility on lubricant-impregnated surfaces *Soft Matter*, 2013,9, 1772-1780.

Reviewer #3 (Remarks to the Author):

I am happy with the responses and the changes incorporated in the manuscript. I opine that the manuscript is ready to be published.

Reviewer #2 (Remarks to the Author):

The authors have confirmed that the substrates in their experiments are 'Lubricant Impregnated Rough surfaces'. The original misleading information in abstract remains unchanged in the revised manuscript. The entire work is based on the point of 'pinning-free' substrate. The authors must define clearly what 'pinning' means, as the edge of drop on a lubricant film follows Neumann's triangle related to the surface energies. How is their situation compared to the contact line pinning reported in this work? Droplet mobility on lubricant-impregnated surfaces Soft Matter, 2013,9, 1772-1780.

We thank the reviewer for his/her further comment on our paper.

As the reviewer correctly points out, the novelty of snap evaporation mode relies on the elimination of pinning effects from a surface. We believe that this is explained with clarity in the following key sentences in the paper.

Abstract:

"While seemingly simple, the configuration of evaporating droplets on solids is difficult to predict and control. This is because evaporation typically proceeds as a "stick-slip" sequence—a combination of pinning and de-pinning events dominated by static friction, or "pinning", caused by microscopic surface roughness. Here we show how smooth, pinning-free, solid surfaces of non-planar topography promote a different process called snap evaporation."

Introduction, last paragraph of page 1 and first paragraph of page 2:

"So far, the widespread conception has been that contact-line pinning caused by microscopic surface roughness dominates the evolution of evaporating droplets."

"Here we show that droplets evaporating on a smooth—but non-flat—solid surface exhibit a different mode of evaporation: instead of pinning the droplet in an uncontrolled manner, the underlying smooth topography promotes a reproducible sequence of well-defined droplet configurations paced by dynamic "snap" events."

Discussion, second column of page 6:

"Unlike stick-slip evaporation, the alternation of well-defined quasi-static states observed in snap evaporation is controlled by shape bifurcations of the liquid-gas interface dictated purely by the inter-play between the underlying surface topography and the droplet volume, and not by contact-line pinning."

We thank the reviewer for his/her suggestion of adding a reference to the work of Smith and co-workers. In the Results section of the manuscript we had referred to the recent theoretical work of Semprebon et al., which discusses in detail the lubricant wetting ridge of a droplet on a Lubricant-Impregnated Rough Surface, see reference 19 of the revised manuscript in the second column of page 2:

"We found that the squared base radius of the droplet decreases linearly with time, while the apparent contact angle, θ_a (measured relative to the horizontal), decreases smoothly due to the effect of the wetting lubricant ridge [18,19]."

We believe that a reference to an experimental paper is appropriate, and have added a citation of Smith et al. alongside (new reference 18 of the revised manuscript).